# PET/CT Imaging and Physiology of Mice on High Protein Diet

**DOI:** 10.3390/ijms22063236

**Published:** 2021-03-22

**Authors:** Jürgen W. A. Sijbesma, Aren van Waarde, Lars Stegger, Rudi A. J. O. Dierckx, Hendrikus H. Boersma, Riemer H. J. A. Slart

**Affiliations:** 1Department of Nuclear Medicine and Molecular Imaging, University Medical Center Groningen, University of Groningen, Hanzeplein 1, 9713GZ Groningen, The Netherlands; a.van.waarde@umcg.nl (A.v.W.); r.a.dierckx@umcg.nl (R.A.J.O.D.); h.h.boersma@umcg.nl (H.H.B.); r.h.j.a.slart@umcg.nl (R.H.J.A.S.); 2Department of Nuclear Medicine, University Hospital, University of Münster, Albert-Schweitzer-Campus 1, 48149 Münster, Germany; stegger@ukmuenster.de; 3Department of Clinical Pharmacy and Pharmacology, University Medical Center Groningen, University of Groningen, Hanzeplein 1, 9713GZ Groningen, The Netherlands; 4Department of Biomedical Photonic Imaging, Faculty of Science and Technology, University of Twente, Drienerlolaan 5, 7522NB Enschede, The Netherlands

**Keywords:** high protein diet, myocardial [^18^F]FDG uptake, myocardial volume, mice

## Abstract

Background: High protein (HP) diets have been proposed to reduce body weight in humans. The diets are known to alter energy metabolism, which can affect the quality of [^18^F]FDG PET heart images. In this preclinical study, we therefore explore the impact of a prolonged HP diet on myocardial [^18^F]FDG uptake. Methods: C57BL/6J (Black six (Bl6)) and apolipoprotein E-deficient (*apoE*^−/−^) mice were fed chow, a HP diet, or a low protein (LP) diet for 12 weeks. At baseline and after treatment, the animals were injected with 33.0 MBq of [^18^F]FDG and a 30 min PET/CT scan was made. Myocardial volume and [^18^F]FDG uptake were quantified using PET and the % of body fat was calculated from CT. Results: Myocardial [^18^F]FDG uptake was similar for all diets at the follow-up scan but an increase between baseline and follow-up scans was noticed in the LP groups. Myocardial volume was significantly smaller in the C57BL HP group compared to the other Bl6 groups. Body weight increased less in the two HP groups compared to the chow and LP groups. Body fat percentage was significantly higher in the LP groups. This effect was stronger in C57BL mice (28.7%) compared to *apoE*^−/−^ mice (15.1%). Conclusions: Myocardial uptake of [^18^F]FDG in mice is not affected by increased protein intake but [^18^F]FDG uptake increases when the amount of protein is lowered. A lower body weight and percentage of body fat were noticed when applying a HP diet.

## 1. Introduction

A high protein diet is an increasingly popular strategy to treat obesity and to change body composition. Protein diets have a high satiety value, leading to changes in appetitive signaling and a reduced energy intake, and, consecutively to loss of body weight [1,2]. Replacing carbohydrates with proteins can result in an increased activation of the amino acid metabolic pathway which is potentially beneficial to prevent or suppress health problems like neonatal morbidity and mortality, diabetes, and cardiovascular disease [3].

[^18^F]2-fluoro-2-deoxy-D-glucose ([^18^F]FDG) positron emission tomography (PET) imaging is a valuable tool to assess cardiac (dys)function and pathology.

In order to detect cardiac inflammation and infection, myocardial [^18^F]FDG uptake should be suppressed to make a distinction between pathologic and physiologic uptake possible [4]. Reduced [^18^F]FDG uptake is acquired by shifting from a glucose-based metabolism to a fatty-acid-based metabolism, for example by feeding a ketogenic diet [5].

For assessment of myocardial viability, an increased [^18^F]FDG uptake is demanded. High [^18^F]FDG uptake is then an indication for myocardial viability, whereas decreased [^18^F]FDG uptake is a sign of myocardial damage, e.g., caused by infarction [6].

For most studies using [^18^F]FDG, elevated insulin levels are essential, attained either by glucose loading (oral glucose administration) or by direct insulin administration (e.g., via an insulin clamp procedure). Acipimox, a nicotinic acid derivate, may be an alternative to clamping [7]. Acipimox inhibits peripheral lipolysis and therefore reduces plasma free-fatty acid levels, and, indirectly, stimulates cardiac [^18^F]FDG uptake [8].

Proteins obtained from diet intake are fragmented to amino acids and transported via the bloodstream to different parts of the body to be used for protein synthesis (anabolism). Amino acids can also be used as an energy source when the energy supplied by carbohydrates and fatty acids is insufficient. The amino acid is deaminated and the resulting carbon skeleton can be oxidized [9].

Activation of the amino acid metabolic pathway is a potential risk factor for impaired quality of [^18^F]FDG PET scans of the human heart. The effect of triggering this pathway on the quality of myocardial [^18^F]FDG scans is unknown. For this reason, we explored the impact of a prolonged high protein diet on the myocardial uptake of [^18^F]FDG, and other relevant physiological parameters, such as myocardial volume, body weight, and the percentage of body fat.

## 2. Results

### 2.1. Effects of Diets on Myocardial [^18^F]FDG Uptake

When comparing the average SUV mean from the chow, LP, and HP groups at the follow-up scan, no significant differences in myocardial [^18^F]FDG uptake were found (Figure 1). However, a significant effect was noticed between strains and between baseline and follow-up scans.

Myocardial SUV was on average 13.6% higher (*p* = 0.023) in *apoE*^−/−^ mice (1.76) compared to Bl6 mice (1.52). Using the Generalized Estimating Equations (GEE) to analyze the strains at each time point, a difference between the strains was only noticed in the follow-up scan (*p* = 0.001).

Follow-up scans (1.79) overall demonstrated a 21.6% higher [^18^F]FDG uptake (*p* < 0.001) compared to baseline scans (1.41).

When looking in more detail at the GEE analysis (per scan, per strain, per diet) we noticed in both the Bl6 LP and *apoE*^−/−^ LP groups a significantly higher myocardial uptake at the follow-up scan. This effect was not noticed in the other groups. Glucose correction of the SUV data was performed (values not shown), but the data and the study outcome were unaffected.

### 2.2. Effects of Diets on Blood Glucose Levels

Blood glucose levels (Table 1) showed no significant difference between diets but we found a difference between the strains (*p* < 0.001) with 15.6% higher levels for the *apoE*^−/−^ mice.

### 2.3. Effects of Diets on Myocardial Volume

The myocardial volume showed an overall significant difference between the diets (*p* = 0.002) with an average volume of 0.047 cm^3^ for animals treated with the HP diet, 0.043 cm^3^ for animals treated with the LP diet, and 0.054 cm^3^ for animals treated with the chow diet.

When applying the GEE analysis, differences in volume were only noticed between diets in the Bl6 groups (Figure 2) with a significant difference between HP and LP (*p* = 0.029), HP and chow (*p* = 0.001), and LP and chow (*p* < 0.001).

In the *apoE*^−/−^ groups, we noticed an increased myocardial volume after treatment compared to baseline in the HP (*p* = 0.003) and LP (*p* < 0.001) groups.

### 2.4. Effects of Diets on Body Weight and Body Composition

All animals in all groups significantly gained body weight (*p* < 0.001) between the start and the end of the diet treatment (Figure 3). In the Bl6 mice, the increase of body weight was significantly smaller in the HP group compared to the chow (*p* = 0.042) or the LP groups (*p* < 0.001). The *apoE*^−/−^ mice demonstrated a similar effect with significantly lower body weight in the HP group compared to the chow (*p* < 0.001) or the LP (*p* = 0.001) groups. The body weight was significantly higher in the Bl6 LP group than in the Bl6 chow group or the *apoE*^−/−^ LP group.

Besides an overall time and diet effect, the GEE indicated an overall strain effect, Bl6 mice being 8% (*p* < 0.001) heavier than *apoE*^−/−^ mice.

In both strains, the LP groups demonstrated a significantly higher percentage of body fat compared to the HP or the chow groups (Figure 4). Although the percentage of body fat was lowest in the HP groups, it was not significantly different from the chow groups. The Bl6 HP and Bl6 LP groups showed a significantly higher fat percentage compared to the *apoE*^−/−^ HP or the *apoE*^−/−^ LP group. Besides a diet effect, a strain effect was also noticed with significantly higher percentages of fat for the Bl6 mice compared to the *apoE*^−/−^ mice.

Figure 5 demonstrates a strong correlation between body weight and the percentage of body fat in the Bl6 groups. The relationship between the two parameters was highly significant for these groups. In the *apoE*^−/−^ mice a significant relationship between body weight and percentage of body fat was only observed in the LP group, with a strong correlation.

## 3. Discussion

This study was performed to explore the effect of a prolonged high protein diet on myocardial [^18^F]FDG uptake and other parameters, such as plasma glucose levels, myocardial volume, body weight, and the percentage of body fat. We demonstrate that the HP diet has no impact on the physiological myocardial uptake of [^18^F]FDG in mice (Figure 1). However, we notice an increase in myocardial uptake between baseline and follow-up scan in both the Bl6 LP and the *apoE*^−/−^ LP group. This effect was not noticed in the follow-up scan because the baseline of the LP was, most likely because of seasonal effects, (significantly) lower compared to the baseline of the HP groups.

The importance of this finding is that in future studies involving an increased protein intake, the physiological myocardial uptake of [^18^F]FDG will be stable and will not be affected by increased protein intake. However, caution is advised because lowering the protein intake in favor of carbohydrates increases myocardial uptake. For this reason it would be of interest to see what happens when protein intake is reduced or increased in favor of fatty acids. Although the HP diets did not seem to affect myocardial [^18^F]FDG uptake we saw a clear diet effect in the myocardial volume (Figure 2), body weight (Figure 3) and percentage of body fat (Figure 4).

Previous papers indicated that a HP diet can contribute to loss in body weight and changes of body composition in humans [10,11] and experimental animals [12,13]. In line with those studies, the current results demonstrate that a HP diet significantly slows down the increase in body weight compared to the chow and LP diet.

HP diet seems to not affect the percentage of body fat, unlike the LP diet which significantly increases the percentage of body fat. Although we did not find any significant differences in the percentage of body fat between HP and chow groups, a clear trend was noticed with, on average, the lowest percentage of body fat in the two HP groups.

Our study focused on the commonly used healthy Bl6 mice and the apolipoprotein E-deficient mice, a model for atherosclerosis, which have a Bl6 background. Besides a diet effect, we found significant differences in myocardial uptake between both strains which suggest that myocardial uptake can be affected by strain differences, but more likely by the pathophysiological processes. Thus, it is of interest to explore the impact of diets on myocardial uptake in other mouse strains and disease models, for example, obesity or diabetes models. Likewise it is of interest to explore the effect of a HP diet with other metabolic tracers than [^18^F]FDG, such as the fatty acid tracers FTHA and BMIPP.

Besides strain differences in myocardial uptake, we also found strain differences in myocardial volume, body weight, and percentage of body fat. We could not find a correlation between volume and body weight and volume and percentage of body weight but we found a correlation between body weight and percentage body fat (Figure 5).

The relationship between the two parameters suggests that an increase in body weight above 30 g is mainly caused by an increase in adipose tissue. Since the body weight of *apoE*^−/−^ mice was lower or around 30 g it is expected to find lower levels of body fat in the *apoE*^−/−^ mice. The lower body weight of the *apoE*^−/−^ mice could be explained by lower body weight at the start of the experiment and a slower increase in body weight [14] caused by the pathophysiological processes in the model.

Animal studies offer the possibility to control different parameters like diet content and intake but often require the use of anesthetics (especially in PET imaging). Isoflurane anesthesia stimulates glucose consumption which can obscure suppressive effects of heparin, fasting, and possibly, (a HP) diet [15]. In order to minimize the impact of isoflurane, we carried out a relatively short static scan of 30 min at 3 h after tracer injection. During the interval between injection and PET scanning, the animals were awake. To reduce the impact of isoflurane exposure even more in future studies, it is possible to perform the [^18^F]FDG injection in a conscious animal by using a restrainer or to perform an ip injection.

A limitation in every diet study is the ratio between fat, carbohydrates, and protein. Changing protein levels in a diet simultaneously changes the levels of carbohydrates and/or fat. Like previous studies concerning the effect of HP diets on vascular parameters and atherosclerosis in mice, we chose similar isocaloric HP and LP diets without changing the levels of fat between the diets [16,17,18]. Although in the HP diet the energy in kcal/kg supplied by protein is higher compared to the chow and LP diets, it is still possible that the available energy in fat in the HP diet blocks a shift to the less favorable amino acid metabolic pathway for energy in the myocardium, since fatty acids and glucose are the main energy sources for the heart [19,20].

According to the Institute of Medicine (2005), the recommended daily intake (RDI) for protein in humans is 0.8 g protein/kg body weight [21,22]. Acceptable Macronutrient Distribution Ranges (AMDR) for protein is between 10 and 35% of the total energy intake [21,22]. Diets with intake values over 35% can be considered as high protein diets [22] since they contain more than 4 times the RDI for protein for an average person [23]. The minimal recommended protein per kg of food is 180 g/kg for mice [24]. The amount of available protein in the HP diet is 497.9 g/kg (Table 2) which is almost 3 times higher than what is recommended. If we translate the human recommendations directly to the animal diets, we can see that the amount of protein in our HP diets is within the AMDR but can be increased without causing a nutrient deficiency. This is possible by for example lowering the amount of fat so that it comes closer to the minimum recommended value of 50 g/kg [24].

Based on this limitation and the possibility to increase the protein levels in the diet without causing a lipid deficiency, we suggest a pilot study to estimate the ratios between protein, carbohydrates, and fat required to switch metabolism to the amino acid metabolic pathway.

## 4. Methods

### 4.1. Animals

Healthy male C57BL/6J (Bl6) and apolipoprotein E-deficient (*apoE*^−/−^) mice (Jackson Laboratory, Bar Harbor, ME, USA, a model for atherosclerosis with associated high risk for various cardiovascular diseases, such as stroke and myocardial dysfunction), with an age of 12 ± 2 weeks were included in this study. All animals were fed ad libitum with a specific diet for a maximum of 12 weeks and were housed at similar conditions, at a 12 h light/12 h dark regime. The study described in this manuscript complied with the Law on Animal Experiments of The Netherlands and was based on approved experimental protocols (DEC 5936A and B).

### 4.2. PET/CT Procedure

On the day of the scan, body weight was measured and non-fasted animals were anesthetized for 8 ± 2 min with a mixture of oxygen (95%) and isoflurane (5% for induction and ≤ 2% for maintenance). A small drop of blood was collected for measuring blood glucose levels. Animals then received an intravenous bolus injection of 33.0 ± 3.8 MBq of [^18^F]FDG. They were placed back in a heated home cage where they could recover from the injection and anesthesia. In order to optimize the contrast between myocardium, the rest of the heart and background, and to minimize the effect of anesthesia during tracer uptake, the mice were again anesthetized approximately 3 h after the injection, and were positioned in a Focus 220 small animal PET camera (Siemens/Concorde Medical Solutions, New York, NY, USA) with the heart at the center of the field of view. A transmission scan (515 s) was made for the correction of scatter and attenuation, using a ^57^Co point source. Exactly 3 h after tracer injection, a 30 min static PET scan was made, followed by a 10 min CT scan (MicroCAT II, CTI Siemens, Munich, Germany) using the following parameters: 60 kV, 360 µA, total rotation of 360° in 500 steps and an exposure time of 1050 ms. The CT scan was used for the measurement of body composition. During all scans, animals were heated using heating pads and electronic temperature controllers (M2M Imaging, Cleveland, OH, USA) set at a temperature of 38 °C. After the baseline scan, the mice were fed a specific diet for 12 weeks. After the dietary treatment of 12 weeks, a second PET/CT scan was made, using the same protocol as was used for the baseline scan.

### 4.3. Diet

The two strains of animals were treated with one of the following diets: (1) a standard chow (ab diets RMH-B), (2) a purified high protein (HP) diet (ab diets 4022,14) or (3) a purified low protein (LP) (ab diets 4022,15) diet. The HP and LP diets were isocaloric.

This resulted in six groups: Bl6 chow (N = 6), *apoE*^−/−^ chow (N = 6), Bl6 HP (N = 12), *apoE*^−/−^ HP (N = 12), Bl6 LP (N = 9) and *apoE*^−/−^ LP (N = 9).

### 4.4. PET Analysis

PET data were reconstructed using an OSEM2D (ordered subsets expectation maximization) reconstruction algorithm with Fourier rebinning, 4 iterations, and 16 subsets. No filter was applied. The final datasets consisted of 95 slices with a slice thickness of 0.8 mm and an in-plane image matrix of 128 × 128 pixels. The voxel size was 0.47 × 0.47 × 0.80 mm and the linear resolution at the center of the field-of-view was 1.5 mm. Scans were corrected for decay, random coincidences, scatter, and attenuation.

Reconstructed PET images were uploaded in the automated three-dimensional contour detection software from the University of Münster. The horizontal long and vertical long axis were manually defined and a contour detection algorithm was used to calculate the myocardial uptake and myocardial volume (cm^3^) [25,26]. To normalize for body weight, PET data are presented as standardized uptake value (SUV).

### 4.5. CT Analysis

CT data were reconstructed using the Filtered Back Projection (FBP) algorithm, creating images with a maximum voxel size of 0.0956 × 0.0956 × 0.0956 mm. Image analysis was performed using Inveon Research Workplace software (Siemens Preclinical Solutions). Since the total body of the mice was not in the field of view we used a segment of the body to determine the percentage of body fat.

A voxel of interest (VOI) was created from the tip of the sternum until the pelvis bone. Bed, anesthesia tubes, and heating pads were removed from the VOI (Figure 6). Hounsfield units with a range of −1000 to +1000 were selected to determine the total body volume within the VOI. Then Hounsfield units with a range of −200 to −50 were selected to determine the total volume of body fat within the VOI [27]. Both volumes were used to calculate the percentage of body fat (Equation (1)).
(1)$$\%bodyfat=\frac{volumebodyfat}{totalbodyvolume}\times \;100$$

### 4.6. Statistical Analysis

The Generalized Estimating Equations (GEE) model was used to account for the repeated measurements in the design and missing data at baseline. The independent correlation matrix was selected for the analysis, and the Wald test was used to report *p*-values, which were considered statistically significant at *p* < 0.05 without correction for multiple comparisons.

Data in figures and tables represent mean ± SD.

## 5. New Knowledge Gained

This study explores the impact of a prolonged high protein diet on myocardial [^18^F]FDG uptake and other parameters, such as plasma glucose levels, myocardial volume, body weight, and the percentage of body fat in mice. A HP diet does not affect myocardial uptake in contrary to the HP diet but does affect body weight and body fat.

## 6. Conclusions

This study demonstrated that myocardial uptake of [^18^F]FDG in mice is not affected by prolonged high protein intake, but lowering protein intake in favor of carbohydrates increases myocardial uptake. A lower body weight and percentage of body fat were noticed when a HP diet was applied.

This suggests that changing protein intake in future studies should be done with caution. To gain more knowledge on the effect of HP diets, other studies like decreasing or increasing protein intake in combination with fatty acids, studies in a variety of disease models (e.g., in obesity and diabetes), and studies with other metabolic tracers are recommended.

## Figures and Tables

**Figure 1 ijms-22-03236-f001:**
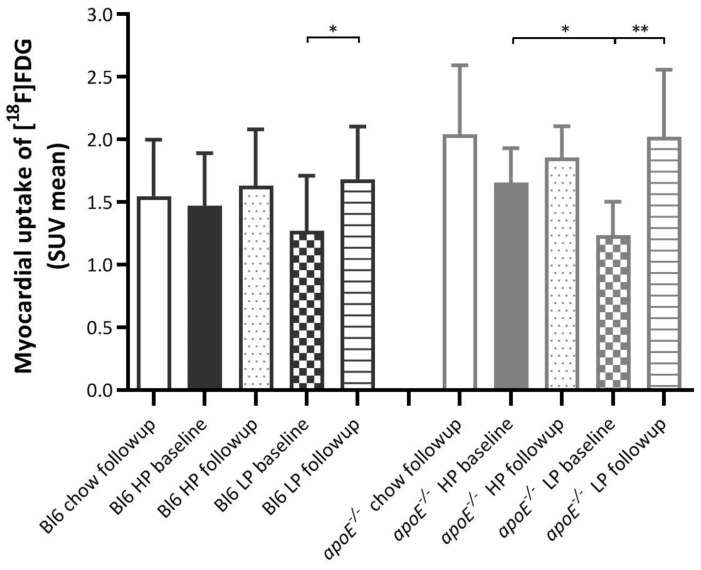
Myocardial uptake (SUV (standardized uptake value) mean) of [^18^F]FDG per group per time point. * is considered significantly different with a *p* < 0.05 and ** significantly different with a *p* < 0.001.

**Figure 2 ijms-22-03236-f002:**
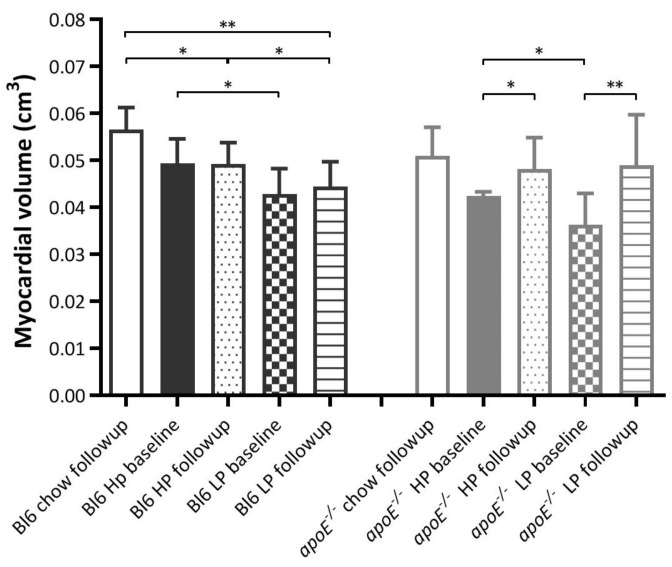
Myocardial volume (cm^3^) per group per time point. * is considered significantly different with a *p* < 0.05 and ** significantly different with a *p* < 0.001.

**Figure 3 ijms-22-03236-f003:**
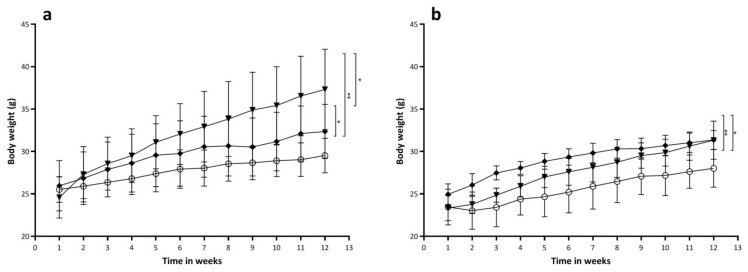
(**a**) Body weight per Bl6 group per time point. (**b**) Body weight per apoE^−/−^ group per time point. Chow, HP (high protein), LP (low protein) diet. Body weight was measured weekly from the start of the diet treatment until the end of the treatment after 12 weeks. * is considered significantly different (*p* < 0.05) and ** significantly different (*p* < 0.001).

**Figure 4 ijms-22-03236-f004:**
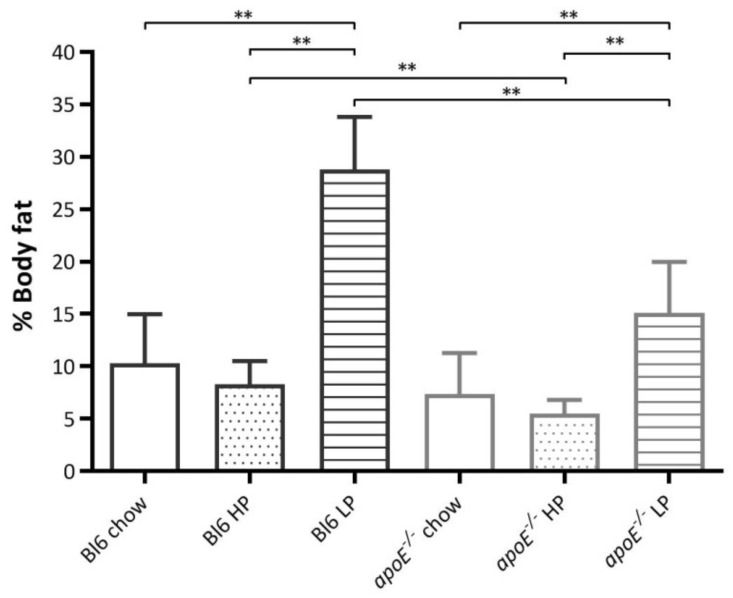
% of body fat per group at the end of the diet treatment. ** is considered significantly different (*p* < 0.001).

**Figure 5 ijms-22-03236-f005:**
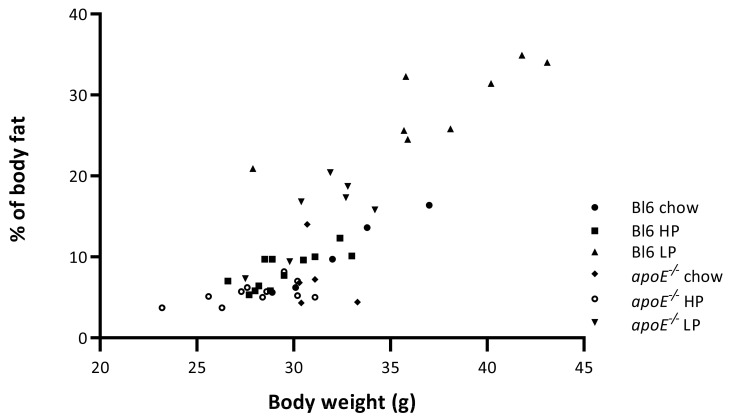
Correlation between body weight and % of body fat per group after treatment. r = 0.985, *p* = 0.002. r = 0.770, *p* = 0.003. r = 0.840, *p* = 0.009. r = 0.336, *p* = 0.580. r = 0.563, *p* = 0.071. r = 0.764. *p* = 0.045.

**Figure 6 ijms-22-03236-f006:**
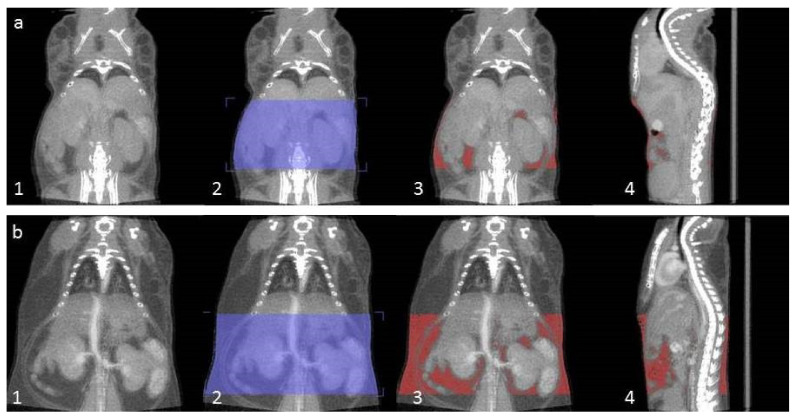
An overview of the steps to determine the % of body fat in bl6 mice treated with (**a**) a hp diet and (**b**) a lp diet. Step 1. Reconstructed ct file is uploaded in research workplace software. Step 2. A voxel of interest (VOI) is drawn from the sternum until the pelvis bone. Within the VOI, Hounsfield units with a range of −1000 to +1000 were selected to determine the total body volume. Hounsfield units with a range of −200 to −50 were selected to determine the volume of body fat within the VOI. Step 3–4. Frontal and sagittal view of the mouse with the body fat shown in red. The volume of adipose tissue in the VOI is used to calculate the percentage of body fat.

**Table 1 ijms-22-03236-t001:** Average blood glucose levels per group and time point.

	Baseline	Follow-Up
Groups	mmol/L	mmol/L
Bl6 chow		8.8 ± 0.8
*apoE*^−/−^ chow		9.3 ± 1.4
Bl6 HP	8.0 ± 1.5	7.8 ± 1.34
*apoE*^−/−^ HP	10.4 ± 1.4	8.5 ± 1.8
Bl6 LP	7.3 ± 0.8	7.8 ± 1.4
*apoE*^−/−^ LP	8.5 ± 1.5	10.1 ± 2.3

**Table 2 ijms-22-03236-t002:** Composition of the chow, high protein, and low protein diet.

	Chow	HP	LP
Gross energy (kcal/kg)	3845.4	4719.4 *	4693.3 *
Ingredients	%	%	%
(Crude) protein	23.5	58.0	27.7
(Crude) fat	5.0	21.0 **	21.0 **
(Crude) fiber	4.3	5.0	5.0
Starch	38.3	1.8	11.4
Sugar	4.0	6.35	27.4
Nutrients	g/kg	g/kg	g/kg
(Crude) protein	-	497.9	239.7
(Crude) fat	-	215.8	212.8
(Crude) fiber	-	49.9	50.9
Sugar + starch	-	85.3	376.1

* HP and LP diets are isocaloric. ** 0.15% cholesterol, - unknown or undefined.

## Data Availability

The PET and CT imaging data reported in this paper are archived in the Groningen facility for Small Animal Imaging (GRONSAI) and are available on request.

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
