# Peer review of "PET/CT Imaging and Physiology of Mice on High Protein Diet"

_ijms, 2021, doi:10.3390/ijms22063236_

Round 1
Reviewer 1 Report
The authors explores the impact of a prolonged high protein diet on myocardial FDG uptake in mice. As a whole, the study is well motivated and the manuscript is clear and easy to read.
major comments
- As a major concern, I really do not see the point of calculating SUV corrected by glucose level. I have been unable to find this equation in the literature. Besides, according to Figure 3, results seems to be unaffected by this correction. So, I understand that this correction does not provide any additional information. Additionally, according to a systematic review published by Webb et al. (2015) glucose level does not affect cardiac metabolism. I understand that glucose level was different among groups and time-points, but this is also true for body weight or body composition and these two factors were not corrected in SUV calculations. This section does not improve the manuscript as it lacks a minimum scientific rigor. The authors should remove these results or, at least, further justify the methodology with references.
- Regarding the results observed in PET, authors should better specify what they are comparing to extract each statement in the first paragraph of results. Particularly:
- Page 5, line 152: “Overall, no differences in myocardial 18F FDG uptake (SUV mean and SUV corr) (figure 3) between the three types of diets were found”. Please, specify which data are compared here, follow-up data?
- In the apoE-/- group, we observe differences at baseline despite the different diet has not yet started. Why?
- In the apoE-/- group, if there are differences in the baseline PET and different diets lead to the same uptake at follow-up, I understand that diet has had an impact on myocardial uptake. Is this inference correct? Please, clarify and/or discuss.
- Page 5, line 161: “we noticed in both the Bl6 LP and apoE-/- LP groups a significantly higher myocardial uptake at the follow-up scan. This effect was not noticed in the other groups”. This sentence involves that there is influence of the diet in the myocardial uptake, at least, with low protein diet and, therefore goes against the first sentence in this section (there is no difference between the three diets).
minor comments
- Regarding the structure of the paper, I wonder if the title is suitable. I would suggest to re-organize the article in ’PET analysis’ and ‘Other variable’, at least if the authors want to give PET analysis the importance that the title suggests. Otherwise, the title should refer to all variables: PET, CT, glucose level, body composition….
- Page 2: “fragmentedto” by “fragmented to”
- Page 2: “contrast between myocardium, the rest of the heart…”. Add a comma
- Page 2, line 66: Authors should include the number of animals in each group.
- Page 3, line 104: Did the authors applied any filter in PET reconstruction? Please, specify the filter or state: “No filter was applied”.
- Page 5, line 148: I do not see the point of using SEM instead of SD, unless authors want to hide a great variability, which, in fact, is common in experimental studies.
Author Response
RESPONSE TO REVIEWER 1
Thank you for the comments and suggestions to improve the manuscript. Kindly find our response to your comments below
- As a major concern, I really do not see the point of calculating SUV corrected by glucose level. I have been unable to find this equation in the literature. Besides, according to Figure 3, results seems to be unaffected by this correction. So, I understand that this correction does not provide any additional information. Additionally, according to a systematic review published by Webb et al. (2015) glucose level does not affect cardiac metabolism. I understand that glucose level was different among groups and time-points, but this is also true for body weight or body composition and these two factors were not corrected in SUV calculations. This section does not improve the manuscript as it lacks a minimum scientific rigor. The authors should remove these results or, at least, further justify the methodology with references.
We also doubted the value of glucose corrected SUV data since there were no significant group differences in glucose levels and the SUV data were unaffected after correction. But glucose corrected SUV data are often requested by readers and reviewers. We agree that the data can better be removed from the manuscript. We have now done so and have only added the following sentence to the results section 3.1., line 74-75: “Glucose correction of the SUV data was performed (values not shown), but the data and the study outcome were unaffected.”.
- Regarding the results observed in PET, authors should better specify what they are comparing to extract each statement in the first paragraph of results. Particularly:
Page 5, line 152: “Overall, no differences in myocardial 18F FDG uptake (SUV mean and SUV corr) (figure 3) between the three types of diets were found”. Please, specify which data are compared here, follow-up data?
In order to clarify what we compare we have changed results section 3.1., line 60-62, as follows: “When comparing the average SUV mean from the chow, LP and HP groups at the followup scan no significant differences in myocardial [18F]FDG uptake were found (figure 2).”
- In the apoE-/- group, we observe differences at baseline despite the different diet has not yet started. Why?
We have no clear explanation for this finding. Because we had delivery problems with the diets, groups were scanned at different seasons. This can be a possible reason.We have included the following words in the discussion line 40: “most likely because of seasonal effects“
- In the apoE-/- group, if there are differences in the baseline PET and different diets lead to the same uptake at follow-up, I understand that diet has had an impact on myocardial uptake. Is this inference correct? Please, clarify and/or discuss.
Page 5, line 161: “we noticed in both the Bl6 LP and apoE-/- LP groups a significantly higher myocardial uptake at the follow-up scan. This effect was not noticed in the other groups”. This sentence involves that there is influence of the diet in the myocardial uptake, at least, with low protein diet and, therefore goes against the first sentence in this section (there is no difference between the three diets).
We agree that there is most likely an effect of the LP diet on the myocardial uptake. When comparing the baseline and followup scans of the LP diet we see that myocardial uptake is increased as mentioned in the results. When only the followup data are considered, this effect is not noticed probably because the LP group had (significantly) lower baseline values compared to the baseline values of the HP group. We agree that we should mention and explain this finding in more detail in the manuscript and should adjust our conclusions.
We changed the text of the manuscript at the following places:
- Line 23-24 and line 30-32 of the abstract.
- Line 38-41 and 44-47, of the discussion
- Line 14-17 of new knowledge gained
- And the complete conclusion
- Regarding the structure of the paper, I wonder if the title is suitable. I would suggest to re-organize the article in ’PET analysis’ and ‘Other variable’, at least if the authors want to give PET analysis the importance that the title suggests. Otherwise, the title should refer to all variables: PET, CT, glucose level, body composition….
We changed the title to: PET/CT imaging and physiology of mice on high protein diet
- Page 2: “fragmentedto” by “fragmented to”
Adjusted
- Page 2: “contrast between myocardium, the rest of the heart…”. Add a comma
Adjusted
- Page 2, line 66: Authors should include the number of animals in each group.
Group numbers are mentioned in section 2.3 Diet line 03-04.
- Page 3, line 104: Did the authors applied any filter in PET reconstruction? Please, specify the filter or state: “No filter was applied”.
“No filter was applied” has been included in line 010.
- Page 5, line 148: I do not see the point of using SEM instead of SD, unless authors want to hide a great variability, which, in fact, is common in experimental studies.
To make the tables and graphs more consistent we changed in the graphs SEM into SD.
Reviewer 2 Report
In the present study, Sijbesma et al investigated mice to explore the impact of a prolonged high-protein (HP) diet on myocardial [18F]FDG uptake. Myocardial uptake of [18F]FDG in mice was not affected by increased protein intake, when compared to a low-protein (LP) diet. However, a diet effect was seen in the myocardial volume, body weight and body fat.
This is an interesting paper. Some comments:
- Myocardial FDG uptake can be significantly bias by an iv injection requiring isoflurane anesthesia. The authors should have considered ip injection allowing to reduce the time frame of anesthesia. The exact duration of anesthesia would be of interest, if available. Indeed, the isoflurane protocol was shortened by the authors as described in the Discussion, but an ip injection could be even further reduce the duration of isoflurane exposure. This should be discussed.
- SUVs are not commonly used in preclinical studies. %ID/g would be of value. If available, can the authors provide such data?
- Fig. 1 is interesting, but can be removed as it does not add further information. All the relevant information can also be put in the Material and Methods Section. If the authors would like to keep the figure, a scale bar is needed.
- The authors shouold provide representative head-to-head comparisons using PET Images for every strain and every diet, so that the neglibile effect on myocardial uptake is visualized.
- The section on novel knowledge could be prolonged.
- Several fatty acid tracers such as FTHA and BMIPP exist. What do the authors think would be the impact on cardiac uptake when their diet models would be used in the context of such radiotracers?
Author Response
RESPONSE TO REVIEWER 2
Thank you for the comments and suggestions to improve the manuscript. Kindly find our response to your comments below
- Myocardial FDG uptake can be significantly bias by an iv injection requiring isoflurane anesthesia. The authors should have considered ip injection allowing to reduce the time frame of anesthesia. The exact duration of anesthesia would be of interest, if available. Indeed, the isoflurane protocol was shortened by the authors as described in the Discussion, but an ip injection could be even further reduce the duration of isoflurane exposure. This should be discussed.
We agree that an ip injection or an iv injection in a conscious animal can help reduce the exposure to isoflurane anaesthesia (added in the discussion line 84-86). In our case the exposure to isoflurane for the iv injection was about 8 minutes (This information has been added in section 2.2 line 81).
- SUVs are not commonly used in preclinical studies. %ID/g would be of value. If available, can the authors provide such data?
We considered to present our data in %ID/g but we think in our study, standardized uptake value is a better option.
We found big differences in body weight between diet treatment and strains. To be able to compare the different groups we think a normalization of body weight is needed which is included in the standardized uptake value.
Normalization of body weight was not needed if the increase in weight was only caused by body fat (there is no FDG uptake in fat so distribution is hardly affected by an increase in body fat) but our data showed that only an increase in body weight above 30 grams is mainly caused by an increase in fat. Since we also have mice below 30 grams a correction of body weight is needed. To make it clear that we need to normalize the PET data for body weight we added the added: “To normalize for body weight PET data is presented as standardized uptake value (SUV).”(section 2.4. line 18)
- 1 is interesting, but can be removed as it does not add further information. All the relevant information can also be put in the Material and Methods Section. If the authors would like to keep the figure, a scale bar is needed.
Fig. 1 was mainly used to show the type of images acquired after reconstruction of the PET data and after using the software for analyzing uptake in the myocardium. We agree that fig. 1 doesn’t add much extra information. We have therefore decided to remove the images and to place the relevant text in the Material and Methods section.
- The authors shouold provide representative head-to-head comparisons using PET Images for every strain and every diet, so that the neglibile effect on myocardial uptake is visualized.
We removed fig. 1 so no head to head comparison is needed.
- The section on novel knowledge could be prolonged.
This section has been partly rewritten (line 15-17)
- Several fatty acid tracers such as FTHA and BMIPP exist. What do the authors think would be the impact on cardiac uptake when their diet models would be used in the context of such radiotracers?
We have never used the mentioned fatty acid tracers in our research, thus we don’t know which impact a LP or HP diet would have on the cardiac uptake of such radiopharmaceuticals. An exploration of the impact of diet on the uptake of various PET tracers in the heart would be an interesting topic for a follow-up study. We have mentioned this possibility in the discussion (line 65-67) and conclusion (line 27-28)